# Temporal relationship of suicide-related internet searches and suicide rates in Korea: A prewhitened cross-correlation analysis

**Seunghyong Ryu**[ID][1,2], **Honey Kim**[1], **Hee-Ju Kang**[1], **Ju-Yeon Lee**[1], **Jae-Min Kim**[1], **Sung-Wan Kim**[ID][1,3]*

**1** Department of Psychiatry, Chonnam National University Medical School, Gwangju, Republic of Korea, **2** Gwangju Metropolitan Mental Health Welfare Center, Gwangju, Republic of Korea, **3** Mindlink, Gwangju Bukgu Community Mental Health Center, Gwangju, Republic of Korea

* swkim@chonnam.ac.kr

## Abstract

### Objective

Suicide-related internet search patterns may reflect population-level behavioral responses to suicide risk. This study investigated the temporal associations between suicide-related internet search volumes and weekly suicide rates in Korea using rigorous time-series methodologies.

### Methods

Weekly suicide rates and search volumes for 25 suicide-related terms were obtained from national mortality records and Naver DataLab for two distinct periods: 2016–2019 and 2020–2023. Prewhitening was applied to mitigate spurious correlations, followed by cross-correlation analyses to assess temporal relationships at lags ranging from 0 to 8 weeks.

### Results

Searches for "psychiatry" and "workplace stress" demonstrated consistent contemporaneous correlations across both study periods. During the 2020–2023 period, significant contemporaneous associations were also observed for prevention-related terms (e.g., "suicide crisis counseling," "1577-0199," "psychological counseling"), "fatigue," "suicide urges," and "suicide death benefit." In contrast, "depression" exhibited a significant association only during the 2016–2019 period. Across all findings, the observed effect sizes were modest.

**Data availability statement:** All data used in this study are publicly available from third-party sources. 1. Suicide Mortality Data: Available from the Microdata Integrated Service (MDIS) (https://mdis.kostat.go.kr/eng/) by navigating to: Download Service > Health > Cause of Death Statistics > Death_Annual Data_Type A. 2. Population Data: Available from the Korean Statistical Information Service (KOSIS) (https://kosis.kr/eng/) by navigating to: Population > Population statistics based on resident registration > Resident Population by City, County, and District. 3. Internet Search Data: Available from Naver DataLab (https://datalab.naver.com/). The datasets can be reproduced using the search keywords listed in the Methods section and S1 Table.

**Funding:** This research was supported by a grant from the Korean Health Technology R&D Project through the Korea Health Industry Development Institute (KHIDI; grant number: HI22C0219), funded by the Ministry of Health and Welfare, Republic of Korea. The funders had no role in study design, data collection and analysis, decision to publish, or preparation of the manuscript.

**Competing interests:** The authors have declared that no competing interests exist.

## Conclusions

Specific internet search terms, particularly those related to suicide prevention resources and occupational stress, exhibit temporal associations with population-level suicide rates. These findings suggest that monitoring online help-seeking behaviors and work-related stressors could serve as useful indicators for public health planning, although further research is required to elucidate underlying mechanisms and determine practical applications.

## Introduction

As internet use has become ubiquitous, individuals at imminent risk of suicide may increasingly seek suicide-related information online immediately before an attempt or death [1,2]. Exposure to online content regarding suicide methods, pro-suicide websites, and related topics can influence behavior and heighten risk among vulnerable individuals [3,4]. Accordingly, examination of suicide-related search behaviors may yield important insights into the patterns and determinants of suicidal behavior [5,6].

Recent research has examined associations between suicide rates and search volumes for suicide-related keywords on major search engines, primarily using time-series methods [7,8]. Searches for terms broadly related to suicide (e.g., "suicide," "self-harm," "depression") and for method-related terms (e.g., "poisoning," "suffocation," "jumping") have shown significant temporal associations with suicide rates at contemporaneous or lagged intervals [9–11]. However, the consistency of these findings is limited, in part due to methodological challenges, including susceptibility to spurious correlations [12,13]. These limitations underscore the need for more rigorous statistical approaches.

Suicide remains a major public health concern in Korea, with annual rates around 25 per 100,000, approximately twice the Organisation for Economic Co-operation and Development (OECD) average [14]. This elevated burden highlights the need to clarify risk factors and identify effective intervention points in the Korean context [15]. Korea's high level of internet penetration provides an opportunity to investigate suicide-related behaviors through analyses of digital search patterns [16]. While several studies have examined suicide-related search behaviors among adolescents, evidence for the general population remains limited [17,18]. Broadening the focus to the general population may yield actionable insights to inform strategies addressing Korea's persistently high suicide mortality.

In this study, we investigated the temporal relationship between national suicide rates and suicide-related internet search volumes in Korea using cross-correlation analysis. To better capture short-term associations, we analyzed weekly time-series data and performed separate analyses for the periods 2016–2019 and 2020–2023 to account for societal and technological changes, including the COVID-19 pandemic and shifts in internet search behavior. We strengthened methodological rigor by applying prewhitening prior to cross-correlation analysis, thereby minimizing the risk of spurious correlations.

## Materials and methods

### Data sources and study population

We analyzed mortality data from the Microdata Integrated Service of Statistics Korea (accessed on 12/12/2025), which included sex, age, date of death, and cause of death for all deaths occurring between 2016 and 2023 (N = 7,298,820) [19]. Weekly suicide deaths (International Classification of Diseases, Tenth Revision (ICD-10) codes X60–X84) were identified from among the 104 causes of death classified by Statistics Korea. During the study period, 53,143 suicide deaths occurred between January 4, 2016, and January 5, 2020, and 53,221 between January 6, 2020, and December 31, 2023, totaling 106,364 deaths. Monthly population data for 2016–2023 were obtained from the Korean Statistical Information Service's resident registration records (accessed on 12/12/2025) [20]. Weekly suicide rates were calculated as the number of suicides per week divided by the corresponding monthly population, multiplied by 100,000.

We systematically selected 25 Korean search terms related to suicide based on prior literature regarding internet searches, the clinical expertise of the research team–comprising psychiatrists and suicide prevention specialists with extensive experience in Korean suicide epidemiology–and the terms' alignment with key dimensions of suicide epidemiology in Korea, including methods, risk factors, and prevention resources [8–13,16–18]. The terms were grouped into five categories: general (suicide, suicide urges, self-harm, suicide note, suicide death benefit); method-related (suicide method, sleep pills, charcoal briquette [charcoal-burning suicide method], jumping, pro-suicide website); reason-related (debt, unemployment, workplace stress, divorce, bullying); prevention-related (suicide crisis counseling, 1577-0199 [national hotline], suicide prevention center, psychiatry, psychological counseling); and symptom-related (depression, anxiety, insomnia, loneliness, fatigue), as detailed in S1 Table. Weekly search volume data were obtained from Naver DataLab (https://datalab.naver.com), the dominant search engine in Korea (accessed on 12/12/2025) [21]. For each of the 25 selected terms, weekly search volumes were retrieved for two distinct periods: January 2016 to December 2019 and January 2020 to December 2023. Data extraction parameters included both mobile and desktop platforms, encompassing all age groups and genders. Naver DataLab provides normalized weekly search volumes, wherein the week with the highest search activity within a given period is assigned a value of 100, and all other values are scaled proportionally.

### Statistical analyses

All analyses were conducted separately for 2016–2019 and 2020–2023. To ensure consistency in data scaling, weekly suicide rates were rescaled within each period to match the normalized scale of weekly search volumes from Naver DataLab (maximum set to 100). Both weekly suicide rates and search volumes were square root–transformed to stabilize variance.

To minimize spurious correlations, prewhitening was applied. The optimal (seasonal) autoregressive integrated moving average ((S)ARIMA) model was first fitted to each search volume (explanatory) series and then applied to both the explanatory and suicide rate (dependent) series. In this approach, the explanatory series is modeled and prewhitened, and the resulting model is then applied to the dependent series; any remaining correlation represents a genuine temporal relationship rather than one arising from shared trends, seasonality, or other time-dependent patterns [12,13]. For each search term, the best-fitting (S)ARIMA model was selected using the auto.arima function in the "forecast" package in R, with both stepwise selection and approximation disabled to maximize accuracy [22]. Model adequacy was assessed using the Box–Ljung test. Using the selected parameters, (S)ARIMA models were fitted to the search volume series with the arima function in the "TSA" package, and the fitted models were then applied to the suicide rate series [23]. The prewhiten function in the "TSA" package was used to jointly filter both series and compute cross-correlation coefficients (CCFs) in a single step. CCFs were calculated for lags ranging from 0 to 8 weeks (restricted to positive lags) to determine whether search volumes coincided with or preceded suicide rates.

Statistical significance was assessed at a Bonferroni-adjusted level ($\alpha = 0.05/50$ for two-sided testing, equivalent to $P < 0.001$), correcting for 25 search terms across two periods. Cross-correlations significant in both periods were considered robust; those significant in only one period were interpreted as period-specific or inconclusive; those not significant in either period were considered non-significant. All analyses were performed using R version 4.4.3 (R Foundation for Statistical Computing, Vienna, Austria).

This study was approved by the Institutional Review Board of Chonnam National University Hospital (IRB No. CNUH-EXP-2025–254). As only publicly available data were analyzed, informed consent was not required.

## Results

The adequacy of all fitted (S)ARIMA models for the 25 suicide-related search terms across five categories was supported by non-significant Box–Ljung test results in both 2016–2019 and 2020–2023 (S2–S6 Tables).

In the general category, "suicide urges" and "suicide death benefit" showed significant positive correlations with suicide rates at lag 0 during 2020–2023 ($r = 0.261$ and $r = 0.303$, respectively), but not in the earlier period (Fig 1). "Suicide," "self-harm," and "suicide note" were not significantly correlated with suicide rates at any lag (0–8 weeks) in either period.

Method-related search terms showed no significant correlations with suicide rates at any lag in either period (Fig 2).

In the reason-related category, only "workplace stress" demonstrated significant positive correlations at lag 0 in both 2016–2019 ($r = 0.232$) and 2020–2023 ($r = 0.241$), indicating a consistent contemporaneous association with suicide rates. "Debt," "unemployment," "divorce," and "bullying" were not significantly correlated in either period (Fig 3).

Among prevention-related terms, "psychiatry" showed significant correlations at lag 0 in both periods (2016–2019: $r = 0.250$; 2020–2023: $r = 0.266$). In addition, during 2020–2023, "suicide crisis counseling" ($r = 0.266$), "1577-0199" ($r = 0.238$), and "psychological counseling" ($r = 0.255$) were also significant at lag 0. "Suicide prevention center" was not significantly correlated in either period (Fig 4).

In the symptom-related category, "depression" was significantly associated with suicide rates at lag 0 in 2016–2019 ($r = 0.303$), but not in 2020–2023. Conversely, "fatigue" was significant at lag 0 in 2020–2023 ($r = 0.391$), representing the highest correlation across all terms and periods. "Anxiety," "insomnia," and "loneliness" were not significantly correlated with suicide rates in either period (Fig 5).

## Discussion

This study investigated the temporal associations between suicide-related internet searches and population-level suicide rates in Korea using a prewhitened cross-correlation approach. Of the 25 search terms examined across two periods, 9 showed statistically significant associations at the population level, with "psychiatry" and "workplace stress" demonstrating robust contemporaneous correlations across both study periods. In the more recent period (2020–2023), additional prevention-related terms and "fatigue" also exhibited significant contemporaneous associations; however, these were period-specific and warrant cautious interpretation. Despite modest effect sizes and the limited number of significant associations, the observed patterns suggest that population-level monitoring of help-seeking behavior and occupational stress may serve as useful indicators of suicide risk from a public health perspective.

Recent studies have examined temporal associations between suicide-related internet search volumes and suicide rates using time-series approaches. Research in Taipei found that searches for "major depression," "suicide," and "domestic violence" coincided with suicide deaths, whereas searches for "divorce" and "complete guide of suicide" preceded increases by one to two months [8]. In England and Wales, searches for "depression," particularly when combined with "help," were associated with suicide rates within the same month, while direct searches for "suicide" or its methods had limited predictive value [9]. Korean adolescent studies reported that spikes in "self-harm" and "dropout" searches most strongly coincided with suicide deaths at zero-day lag [17,18]. To reduce the risk of spurious correlations, recent studies have applied prewhitening in cross-correlation analyses. In a U.S. study, only a small number of terms, such

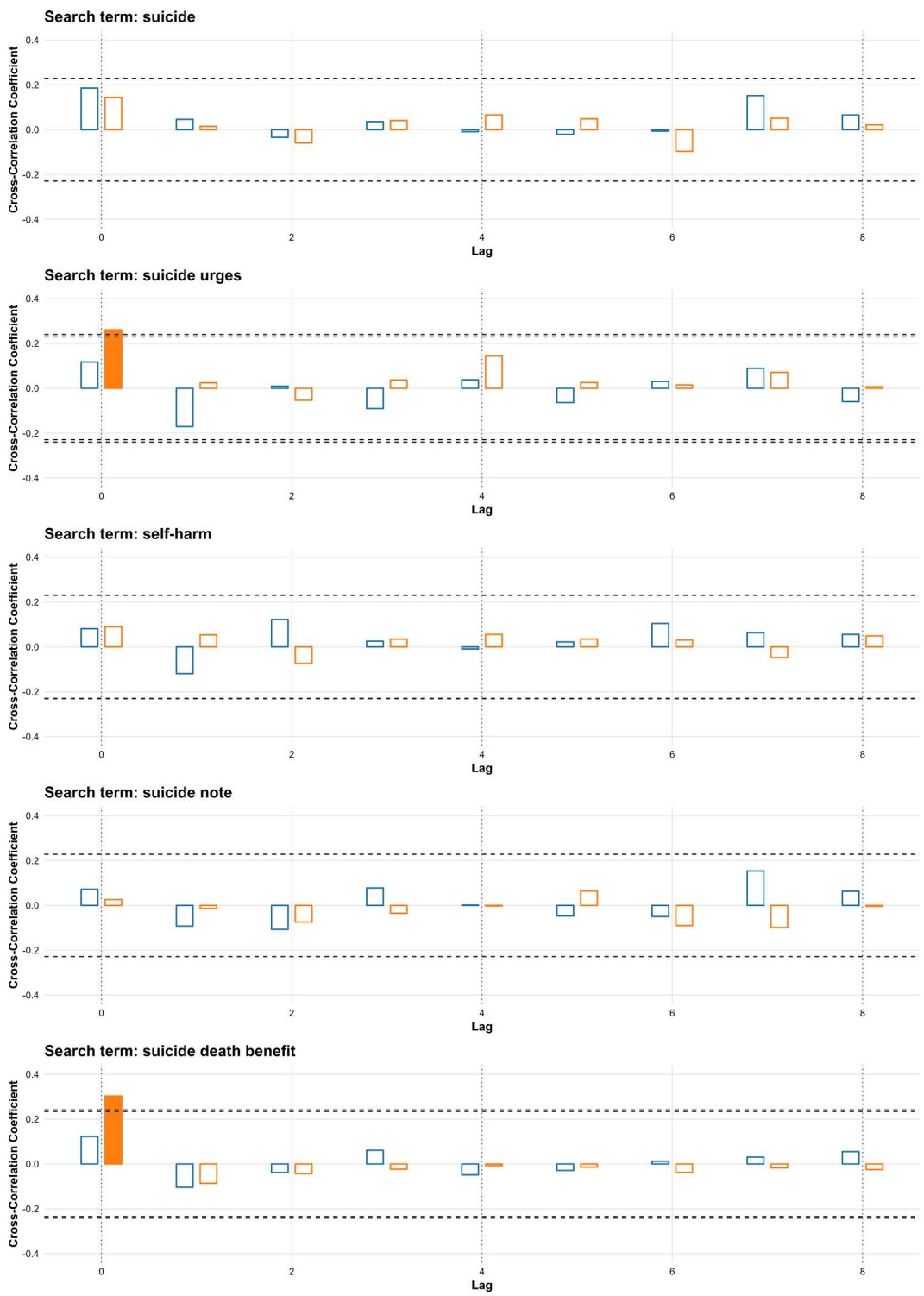

**Fig 1. Cross-correlation between weekly search volumes for general suicide-related terms and suicide rates in South Korea for 2016–2019 and 2020–2023.** Bars represent cross-correlation coefficients for lags of 0–8 weeks; blue bars show results for 2016–2019, and orange bars show results for 2020–2023. Filled bars indicate statistically significant correlations. Horizontal dashed lines denote the threshold for statistical significance.

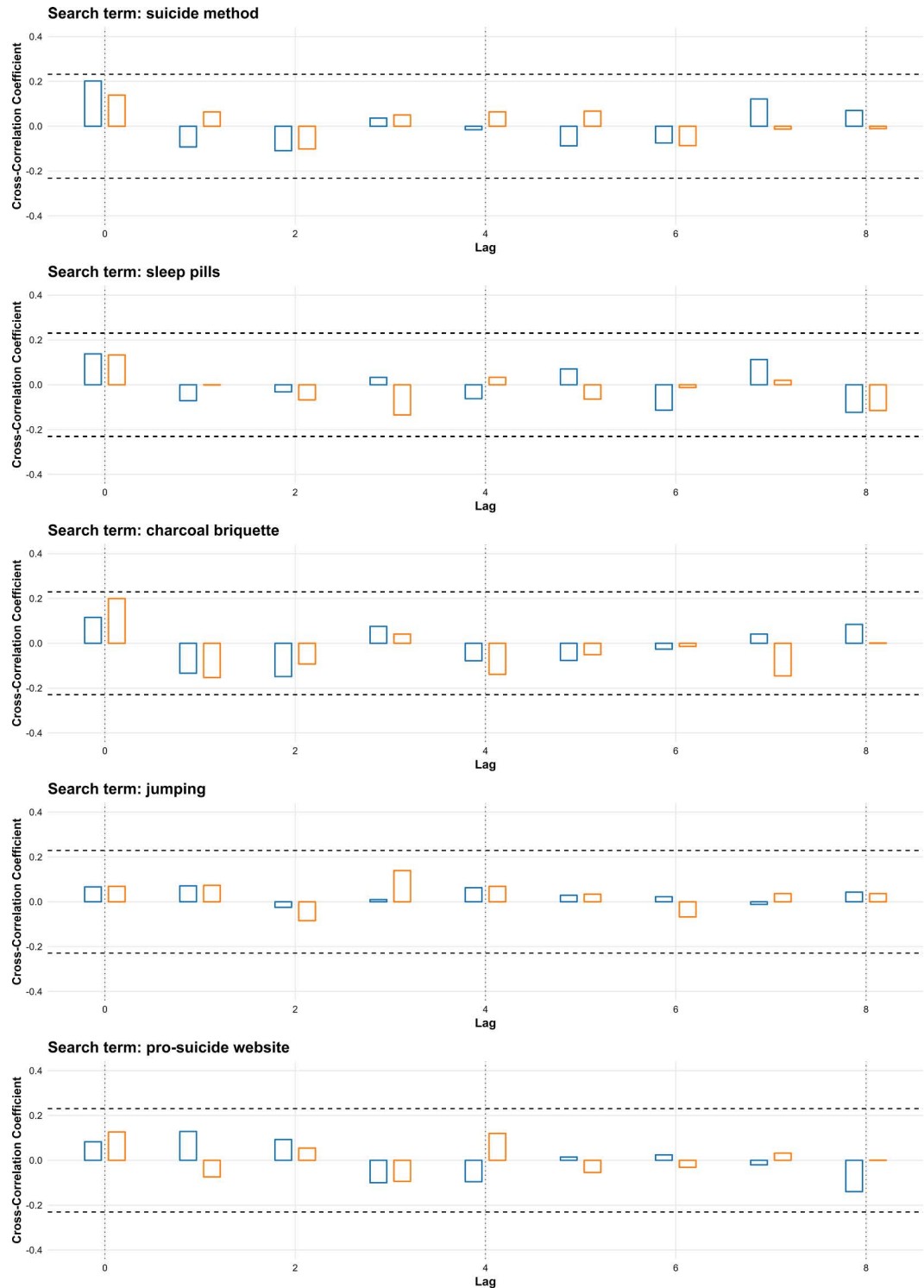

**Fig 2. Cross-correlation between weekly search volumes for method-related terms and suicide rates in South Korea for 2016–2019 and 2020–2023.** Bars represent cross-correlation coefficients for lags of 0–8 weeks; blue bars show results for 2016–2019, and orange bars show results for 2020–2023. Filled bars indicate statistically significant correlations. Horizontal dashed lines denote the threshold for statistical significance.

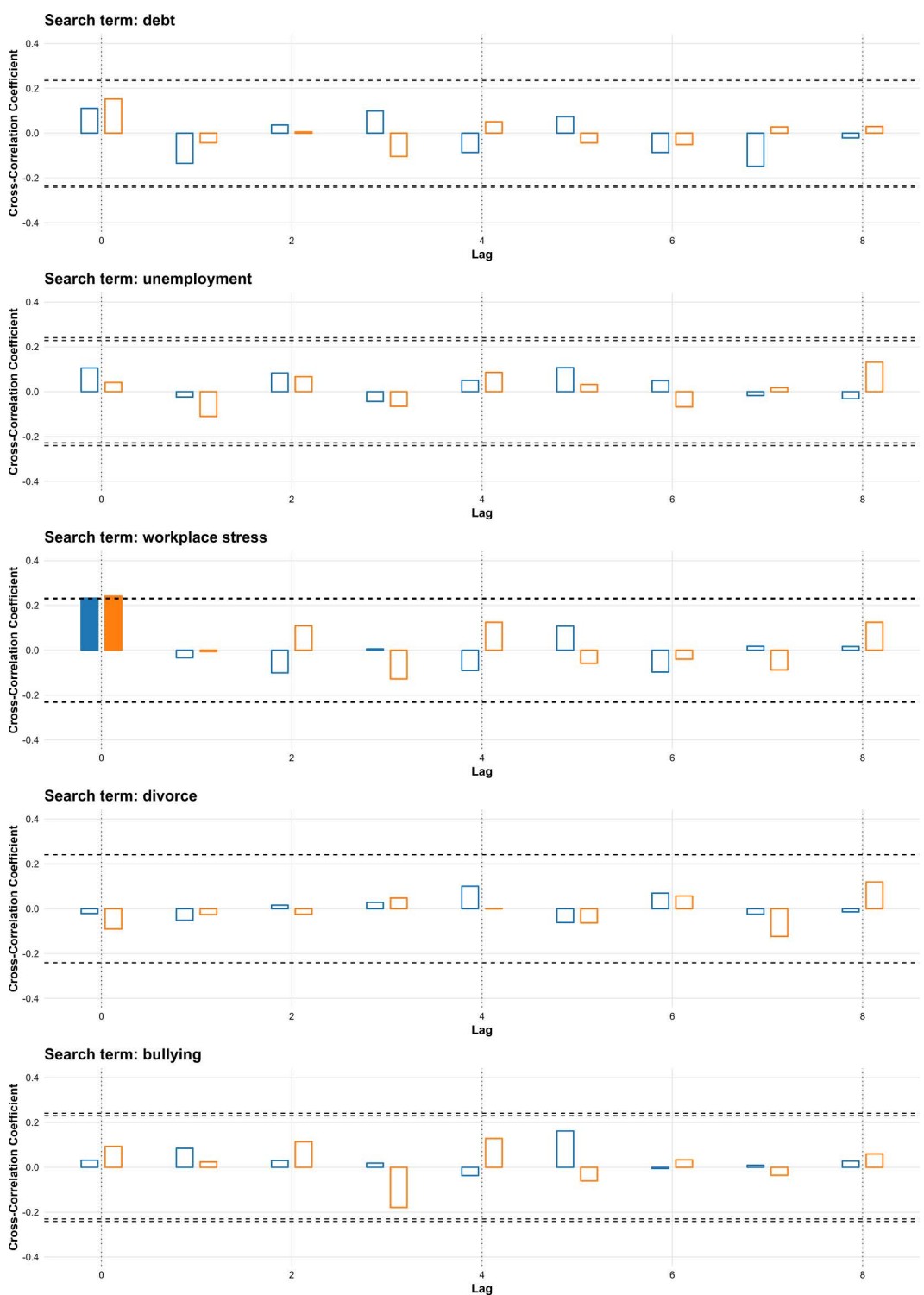

**Fig 3. Cross-correlation between weekly search volumes for reason-related terms and suicide rates in South Korea for 2016–2019 and 2020–2023.** Bars represent cross-correlation coefficients for lags of 0–8 weeks; blue bars show results for 2016–2019, and orange bars show results for 2020–2023. Filled bars indicate statistically significant correlations. Horizontal dashed lines denote the threshold for statistical significance.

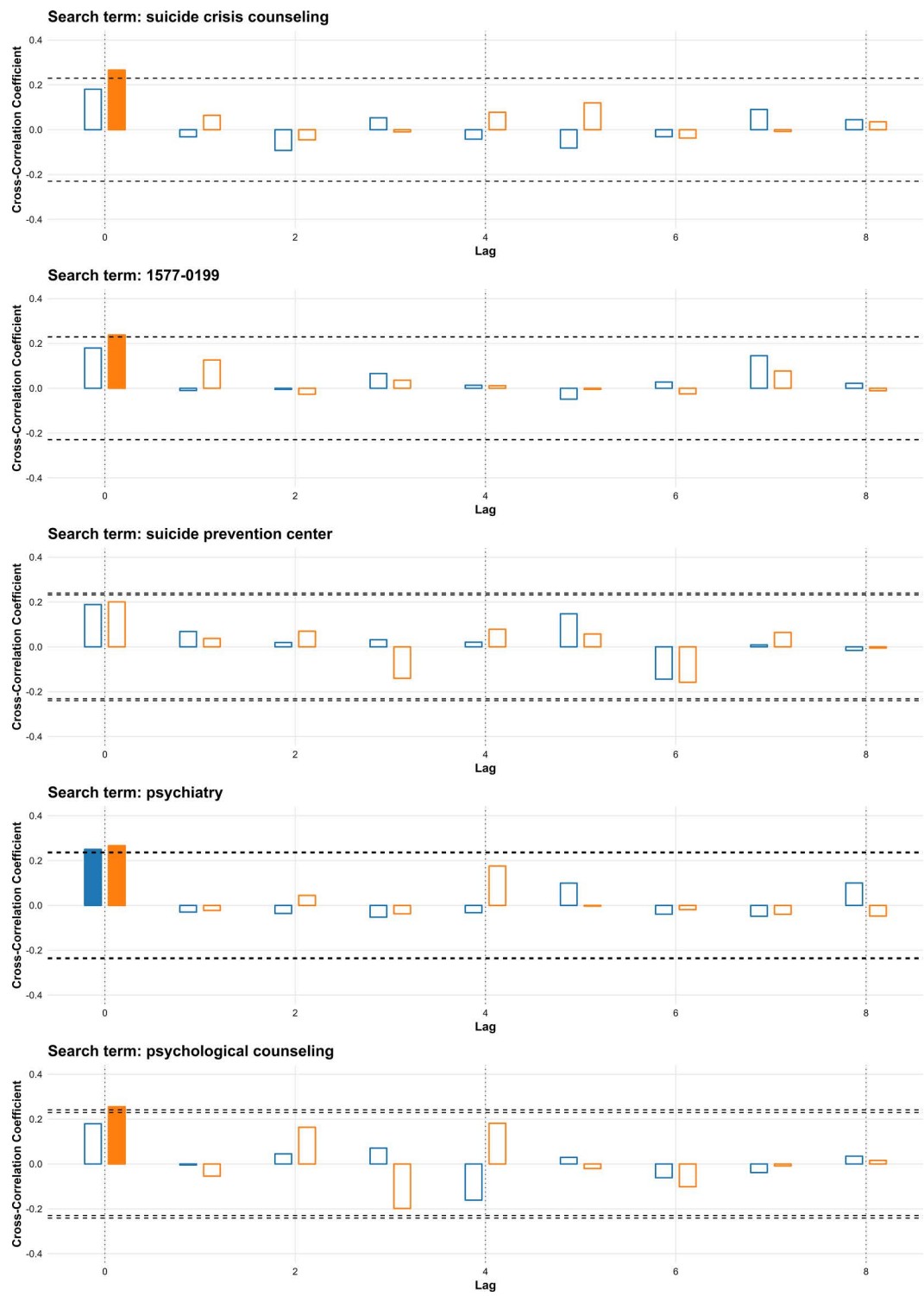

**Fig 4. Cross-correlation between weekly search volumes for prevention-related terms and suicide rates in South Korea for 2016–2019 and 2020–2023.** Bars represent cross-correlation coefficients for lags of 0–8 weeks; blue bars show results for 2016–2019, and orange bars show results for 2020–2023. Filled bars indicate statistically significant correlations. Horizontal dashed lines denote the threshold for statistical significance.

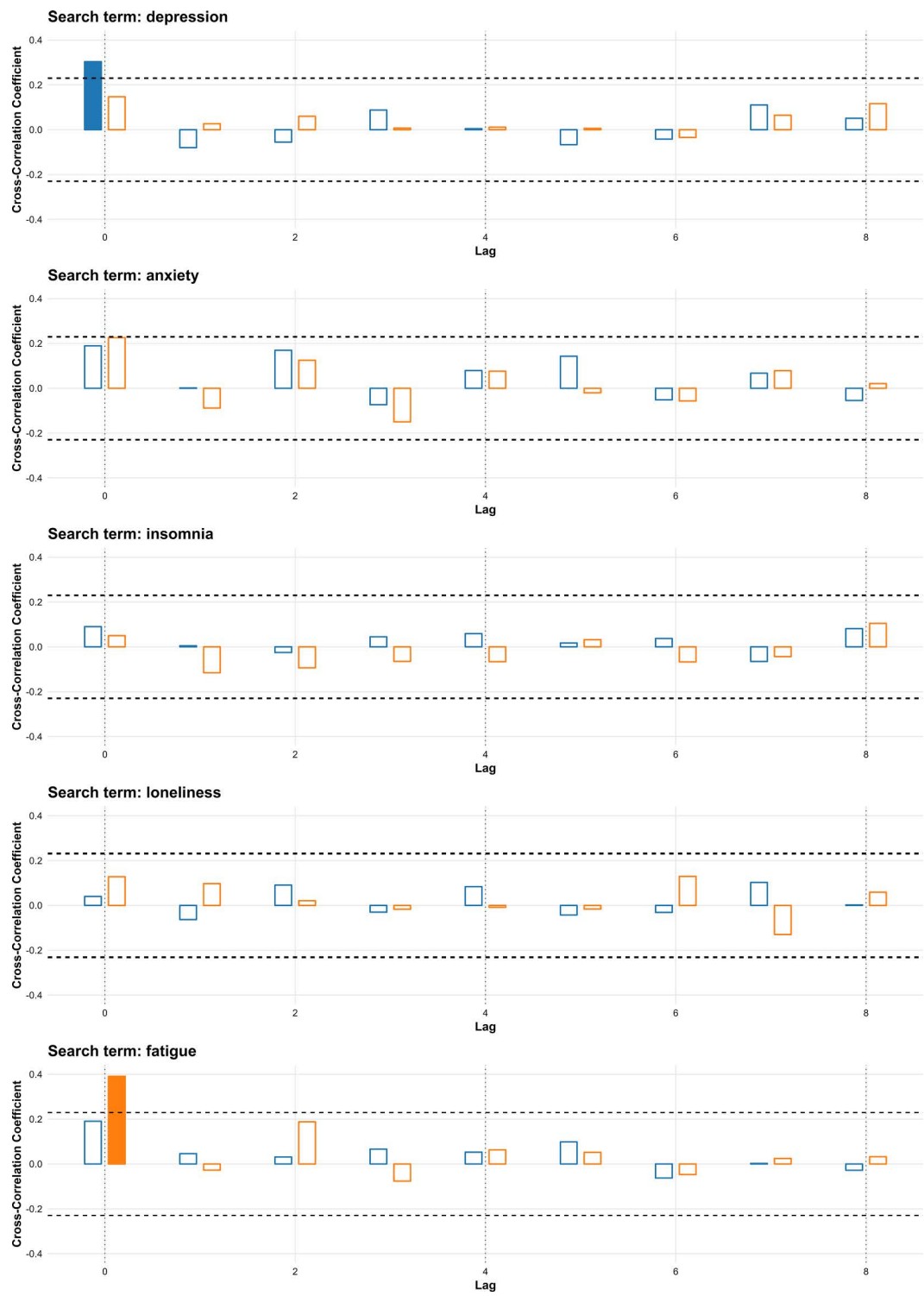

**Fig 5. Cross-correlation between weekly search volumes for symptom-related terms and suicide rates in South Korea for 2016–2019 and 2020–2023.** Bars represent cross-correlation coefficients for lags of 0–8 weeks; blue bars show results for 2016–2019, and orange bars show results for 2020–2023. Filled bars indicate statistically significant correlations. Horizontal dashed lines denote the threshold for statistical significance.

as "generalized anxiety disorder," "anxiety disorder," and "laid off," showed persistent associations with suicide rates, while many traditional suicide and depression terms did not [12]. A cross-national analysis found no robust associations between Google Trends suicide-related search volumes and suicide rates [13]. Collectively, these findings suggest a potential temporal relationship between suicide-related internet search activity and suicide outcomes. However, evidence across studies remains heterogeneous and inconsistent. This inconsistency highlights the necessity of employing rigorous statistical approaches, such as prewhitening, to determine whether robust associations persist and to minimize the spurious correlations.

To clarify the temporal association between suicide rates and suicide-related internet searches, we applied a rigorous, contextually informed methodology. Search terms were systematically selected to capture key aspects of suicide epidemiology in Korea, including common causes, prevalent methods, major prevention resources, and terms examined in previous research. The use of weekly rather than monthly data enabled finer temporal resolution. To mitigate the risk of spurious correlations arising from the pronounced seasonality and non-stationarity inherent in suicide time-series data, prewhitening techniques were implemented, and model adequacy was subsequently verified using the Ljung-Box test. Analyses were performed separately for 2016–2019 and 2020–2023 to account for societal and technological changes, including increased mobile internet use, shifts in search engine preferences, and the impact of the COVID-19 pandemic [24,25]. Multiple testing corrections were applied to enhance the robustness of the results. Taken together, these methodological strengths increase the validity and interpretability of the observed findings.

A key finding of this study was the relatively strong contemporaneous correlation between suicide rates and search volumes for terms related to suicide prevention, particularly when compared to the other search terms examined. Searches for "psychiatry" showed consistent positive correlations with suicide rates in both 2016–2019 and 2020–2023, indicating a stable association over time. In the more recent period, additional prevention-focused terms, such as "suicide crisis counseling," "1577-0199," and "psychological counseling," also became significant, possibly reflecting increased mental health awareness and help-seeking during and after the COVID-19 pandemic. During this time, Korean media more frequently included prevention resources, such as hotline numbers, in suicide-related news coverage, in accordance with updated reporting guidelines [26]. This pattern is consistent with findings from England and Wales, where increased searches for "depression help" and "suicide prevention" were also associated with suicide rates [9]. In contrast to the positive association observed in our study, research conducted in California and Arizona reported that prevention-related searches were followed by a decrease in emergency department visits for suicidality [27]. This finding aligns with evidence from other studies suggesting that effective help-seeking and crisis support can reduce the risk of suicide attempts [28]. In Korea, however, the concurrent rise in suicide rates and prevention-related searches may indicate barriers such as limited awareness, restricted accessibility, and persistent stigma, leading individuals to seek help online without subsequently engaging with available services [29]. Enduring cultural stigma surrounding mental health care may further limit the effectiveness of help-seeking [30]. These findings underscore the need not only to improve access to and effectiveness of crisis and mental health services in Korea, but also to address societal barriers and stigma to maximize the population-level impact of suicide prevention efforts.

Another important finding was the significant positive contemporaneous correlation between suicide rates and search volumes for "workplace stress" in both study periods. International evidence suggests that 10–13% of suicide deaths may be attributable to adverse working conditions [31]. In Korea, although nearly half of suicides occur among the economically active population, only 3–5% are formally classified as work-related, suggesting that occupational stress may contribute to more suicides than officially recognized, likely due to overlap with other mental health or socioeconomic factors [32]. The higher likelihood of workforce members engaging in internet searches may also partly explain the strong association observed. The role of occupational stress is further supported by Korean studies linking burnout and exhaustion, core features of workplace stress, to suicidal ideation among employees [33]. Consistent with this, we observed a positive correlation between searches for "fatigue" and suicide rates in recent years, supporting the interpretation of fatigue as a marker

of occupational exhaustion and a potential risk factor for suicide [34]. Fatigue is recognized as a significant predictor of suicidal ideation not only among individuals experiencing workplace stress but also in the general population [35]. These findings underscore the need for systematic monitoring of occupational stress and fatigue as key indicators of suicide risk in both clinical and community settings. However, the underlying mechanisms by which population-wide changes in searches for "workplace stress" and "fatigue" coincide with increases in suicide rates remain unclear. Furthermore, given the aggregate nature of the data, we cannot determine whether the individuals conducting these searches are the same individuals at elevated risk of suicide.

Several search terms exhibited significant associations in only one of the two study periods. As indicated in our Methods, these period-specific findings should be interpreted as inconclusive rather than robust, as they may be attributable to chance or transient contextual factors rather than stable temporal relationships. "Depression" showed a significant contemporaneous correlation with suicide rates only before 2020, whereas "suicide urges" were significant only after 2020. These period-specific patterns may reflect contextual influences, including the COVID-19 pandemic and broader social changes affecting public attention and internet search behaviors [25]. However, because each term was significant in only one period, these associations may be inconclusive or attributable to chance rather than representing robust temporal relationships. Prior studies have generally reported inconsistent or weak associations between searches for broad, non-specific suicide-related terms (e.g., "suicide," "depression") and suicide rates [12,13]. The emergence of "suicide death benefit" as a significant correlate in the recent period may be related to increased public debate and legislative changes concerning suicide-related insurance in Korea, potentially influencing search behavior [36]. Furthermore, we observed only limited correlations for direct suicide/self-harm terms and method-specific queries, contrary to concerns that such searches might precipitate increases in suicide outcomes; this pattern may partly reflect search engine restrictions on sensitive suicide-related content [37].

All significant correlations between suicide-related search volumes and suicide rates occurred within the same week (lag 0), with no meaningful correlations at longer lags. This finding is consistent with prior research in Korean adolescents, which reported the strongest associations between suicide-related searches and suicide deaths at short temporal intervals, particularly in daily or weekly analyses [17,18]. These results suggest that suicide-related search activity in Korea may intensify immediately before suicidal acts. However, in the present study, the observed correlations were modest (approximately $r = 0.25$), and searches involving explicit suicidality (e.g., "suicide," "self-harm," "suicide note," method-specific terms) were not significantly associated with suicide rates. Accordingly, real-time surveillance based on suicide-related queries may have limited utility for prevention [13]. Instead, the high frequency of help-seeking searches immediately preceding suicide events underscores the importance of improving public awareness, accessibility, and effectiveness of prevention services to ensure timely support for individuals in crisis [5,6].

This study has several limitations that should be considered when interpreting the findings. First, the observed associations represent temporal correlations and do not imply causality. Future research employing alternative methodologies is required to elucidate the direction and nature of these relationships. Second, only a predefined set of search terms was examined based on prior literature and clinical expertise. Consequently, other relevant search behaviors may not have been captured. While our ecological approach utilizing aggregate data differs from individual-level research, future studies involving individuals with lived experience could help identify additional search terms not captured here. Third, the analysis was based on data from a single search engine and aggregated at the population level; findings may differ for other search platforms or specific demographic subgroups. Fourth, while statistically significant associations were observed, effect sizes were modest and temporal stability was limited. Only two terms were consistent across both study periods, whereas the remaining seven were period-specific and thus inconclusive. This lack of consistency warrants caution in interpretation and limits the generalizability of the findings. Finally, as this is an ecological study, the results are subject to the ecological fallacy and cannot be extrapolated to individual-level behavior. The observed associations may reflect shared socioeconomic conditions or media coverage rather than serving as direct indicators of individual suicidal intent, thereby limiting the utility of internet search data for suicide prediction.

This study demonstrated that specific internet searches, particularly those related to suicide prevention resources and occupational stress, exhibit temporal associations with suicide rates in Korea. These findings offer insights into the behavioral patterns and contextual factors associated with suicide risk at the aggregate level. However, given that these associations were modest in magnitude and cannot identify or predict individual suicide risk, the results should be interpreted with caution. While these findings may inform public health planning by enhancing our understanding of suicide dynamics, further research is needed to determine their effective application in prevention strategies.

## Supporting information

**S1 Table. Search terms examined in the study.**
(DOCX)

**S2 Table. Cross-correlation between weekly suicide-related search volumes (category: general) and suicide rates.**
(DOCX)

**S3 Table. Cross-correlation between weekly suicide-related search volumes (category: method) and suicide rates.**
(DOCX)

**S4 Table. Cross-correlation between weekly suicide-related search volumes (category: reason) and suicide rates.**
(DOCX)

**S5 Table. Cross-correlation between weekly suicide-related search volumes (category: prevention) and suicide rates.**
(DOCX)

**S6 Table. Cross-correlation between weekly suicide-related search volumes (category: symptom) and suicide rates.**
(DOCX)

## Author contributions

**Conceptualization:** Seunghyong Ryu, Sung-Wan Kim.

**Data curation:** Seunghyong Ryu, Honey Kim, Hee-Ju Kang.

**Formal analysis:** Seunghyong Ryu.

**Funding acquisition:** Sung-Wan Kim.

**Methodology:** Seunghyong Ryu, Sung-Wan Kim.

**Supervision:** Ju-Yeon Lee, Jae-Min Kim.

**Visualization:** Seunghyong Ryu.

**Writing – original draft:** Seunghyong Ryu, Honey Kim, Sung-Wan Kim.

**Writing – review & editing:** Hee-Ju Kang, Ju-Yeon Lee, Jae-Min Kim.

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
