## [Decision Letter · Decision Letter 0]

29 Oct 2025

Dear Dr. Kim,

Thank you for submitting your manuscript to PLOS ONE. After careful consideration, we feel that it has merit but does not fully meet PLOS ONE’s publication criteria as it currently stands. Therefore, we invite you to submit a revised version of the manuscript that addresses the points raised during the review process.

I agree with both reviewers about the need for further revisions.

We look forward to receiving your revised manuscript.

Kind regards,

Diego A. Forero, MD; PhD

Academic Editor

PLOS ONE

“This research was supported by a grant from the Korean Health Technology R&D Project through the Korea Health Industry Development Institute (KHIDI; grant number: HI22C0219), funded by the Ministry of Health and Welfare, Republic of Korea.”

Reviewers' comments:

Reviewer's Responses to Questions

**Comments to the Author**

1. Is the manuscript technically sound, and do the data support the conclusions?

Reviewer #1: Yes

Reviewer #2: Partly

2. Has the statistical analysis been performed appropriately and rigorously?

Reviewer #1: Yes

Reviewer #2: No

3. Have the authors made all data underlying the findings in their manuscript fully available?

Reviewer #1: No

Reviewer #2: No

4. Is the manuscript presented in an intelligible fashion and written in standard English?

Reviewer #1: Yes

Reviewer #2: Yes

Reviewer #1: This is a well written paper building on existing literature seeking to predict suicide trends using online search behaviour.

I am satisfied with the writing and statistical analysis, and will male some small typographic suggestions, and then one big suggestion about the discussion.

P3L3-4 motivate the two separate periods (covid?). This is only mentioned in the discussion on P9L10-12, which is too late.

P3L21 provide citations for “prior literature”

P12L11 Granger causal analysis, despite the name, can also not show causality. It’s use in this sentence is not right.

Discussion:

P8L7-9 I think is overstating the results, as is P9l15-16.

In the methods, the authors state that "Cross-correlations significant in both periods were considered robust; those significant in only one period were interpreted as period-specific or inconclusive; those not significant in either period

were considered non-significant". However, in the discussion, nothing is treated as inconclusive. Furthermore, of 50 correlations (25 terms across two periods), very few (ten by my count) were significant. Only two terms were what the authors called "robust". I think wholesale acceptance of the premise that search trends can predict suicide rates is really overstating these rather modest results. The authors should rework the whole discussion to be more in keeping with their results, and not "pounce" on some significant findings as proof of the overall hypothesis. And it should be noted that even those that were statistically significant had quite small effect sizes.

There is an even greater caution that should also be recognised. The authors are studying what is essentially population-level data, with essentially individual-specific keywords. While mass suicide is, unfortunately, sometimes a reality, suicide is mostly an individual behaviour. As such, (and as an example), there is a missing link why changes in searches for something like workplace stress should correlate with suicide rates, because it does not address the question of what would lead to increases in workplace stress for large parts of the population all at the same time (so that there would be, for example, an increase in searches for workplace stress), which is then correlated to increases in suicide. Without a proper understanding of this mechanism, attempts to predict suicide through online searches remains, at most, an interesting idea with limited practical application.

Reviewer #2: Thank you for the opportunity to review this paper on the correlation between suicide and suicide-related online searches in Korea. There are two main concerns regarding this paper that I believe should be addressed before the manuscript could be accepted for publication.

1. Overstating claims

In the discussion, many claims and interpretations are overstated. I highlight a few here:

“Collectively, these findings indicate that suicide-related internet search activity often occurs contemporaneously or at short lags relative to suicide rates, underscoring the importance of rigorous statistical approaches, including prewhitening, to ensure validity.” I am not sure how you came to this conclusion?

“Search terms were systematically selected to capture key aspects of suicide epidemiology in Korea, including common causes, prevalent methods, major prevention resources, and terms examined in previous research. “ This is a potential limitation and it needs to be addressed more thoroughly than you currently do in the limitation section. The current opinion in the field of suicidality is that involving people with lived experience is necessary; why has this not been done? Also please provide details on how the search terms were systematically selected.

You mention “strong contemporaneous correlation” Was it? 0.25-0.26 is moderate at best.

“In contrast, a study from California and Arizona reported that prevention-related searches were followed by declines in emergency department visits for suicidality [27].” I do not understand what is that finding in contrast with?

Not sure if prewhitenting is such a revolutionary methodological approach. It has been done in quite a few papers on this topic, most recently Onie at al (2025) and Colbeth et al (2024).

2. Methodological issues

While authors did employ correct statistics, there are a couple more things to address before the approach could be considered rigorous.

a) It is unclear if the Bonferroni correction of p-values was applied to ccf too? Based on the figures, I would guess not.

b) Why did you not examine negative lags? -8 to +8?

c) How were missing values from the Naver Data Lab handled? Were there any missing values?

d) Please include how the Naver Data was obtained, processed, and aggregated. Unless you do so, your claim that the data is publicly available is not true as researchers cannot replicate your analysis.

e) In your introduction you have correctly identified the common focus on adolescents in similar research. I was surprised that you did not follow that up in your analysis? Who were the people who died by suicide in your analysis? Is it possible to delineate the analysis by age, even binary one such as adolescents/adults?

f) Figures are of low quality/resolution

g) NB rescaling suicide data was not necessary from a mathematical standpoint. The CCF can be applied to any two time series, regardless of their original scale, because the function is based on the correlation coefficient, which is not affected by absolute magnitude as long as the series are not constant. Most importantly, such a normalisation may actually obscure the interpretation of your data through the loss of variance or masking outliers/trends. Consider rerunning on actual suicide numbers.

**Do you want your identity to be public for this peer review?** For information about this choice, including consent withdrawal, please see our Privacy Policy

Reviewer #1: **Yes:** Jacques Raubenheimer

Reviewer #2: **Yes:** Lana Bojanić

---

## [Author Response · Author response to Decision Letter 1]

15 Dec 2025

Reviewer #1: This is a well written paper building on existing literature seeking to predict suicide trends using online search behaviour.

I am satisfied with the writing and statistical analysis, and will male some small typographic suggestions, and then one big suggestion about the discussion.

Response: We thank Reviewer #1 for the thorough and insightful review. We have substantially revised the manuscript to address the concerns raised regarding the interpretation of our results and to improve methodological clarity.

P3L3-4 motivate the two separate periods (covid?). This is only mentioned in the discussion on P9L10-12, which is too late.

Response: Thank you for this helpful suggestion. We agree that the rationale for analyzing the two periods separately should be introduced earlier. We have revised the Introduction to clarify that analyses were conducted separately for 2016–2019 and 2020–2023 to account for significant societal and technological changes, including the COVID-19 pandemic and shifts in internet search behavior (highlighted, page 3 lines 2–5).

P3L21 provide citations for “prior literature”

Response: Thank you for pointing this out. We have added appropriate citations to the Methods section to support our statement regarding prior literature on the relationship between internet search behavior and suicide (highlighted, page 3 line 22–page 4 line 2).

P12L11 Granger causal analysis, despite the name, can also not show causality. It’s use in this sentence is not right.

Response: Thank you for the methodological correction. We agree that Granger causality does not establish true causality. We have removed this reference from the Limitations section. The manuscript now acknowledges that our findings reflect temporal correlations rather than causal relationships, and that alternative methodologies are required to clarify the direction and nature of these associations (highlighted, page 13 lines 9–11).

Discussion:

P8L7-9 I think is overstating the results, as is P9l15-16.

In the methods, the authors state that "Cross-correlations significant in both periods were considered robust; those significant in only one period were interpreted as period-specific or inconclusive; those not significant in either period were considered non-significant". However, in the discussion, nothing is treated as inconclusive. Furthermore, of 50 correlations (25 terms across two periods), very few (ten by my count) were significant. Only two terms were what the authors called "robust". I think wholesale acceptance of the premise that search trends can predict suicide rates is really overstating these rather modest results. The authors should rework the whole discussion to be more in keeping with their results, and not "pounce" on some significant findings as proof of the overall hypothesis. And it should be noted that even those that were statistically significant had quite small effect sizes.

Response: We appreciate the reviewer’s critique regarding the interpretation of our findings. We have extensively revised the Discussion to more accurately reflect the modest nature of the results and to avoid overstatement (highlighted, page 8 lines 13–16; page 12 lines 1–4; page 13 lines 17–21).

We explicitly acknowledge that out of 50 tested correlations, only 9 were statistically significant, with merely 2 showing robust associations across both study periods. We have removed any language suggesting strong predictive utility and now frame these findings as exploratory patterns requiring cautious interpretation. Furthermore, we have clarified that associations significant in only one period should be viewed as inconclusive and potentially attributable to chance, rather than representing stable temporal relationships.

There is an even greater caution that should also be recognised. The authors are studying what is essentially population-level data, with essentially individual-specific keywords. While mass suicide is, unfortunately, sometimes a reality, suicide is mostly an individual behaviour. As such, (and as an example), there is a missing link why changes in searches for something like workplace stress should correlate with suicide rates, because it does not address the question of what would lead to increases in workplace stress for large parts of the population all at the same time (so that there would be, for example, an increase in searches for workplace stress), which is then correlated to increases in suicide. Without a proper understanding of this mechanism, attempts to predict suicide through online searches remains, at most, an interesting idea with limited practical application.

Response: We appreciate your critique regarding the inherent limitations of ecological studies. We explicitly acknowledge that our reliance on population-level data prevents us from drawing conclusions about individual suicide risk or the specific mechanisms linking search behavior to suicidal acts.

The revised manuscript now clearly states that observed associations may reflect shared contextual factors, such as socioeconomic conditions or media coverage, rather than direct indicators of individual behavior. We have reframed our conclusions to emphasize that these findings contribute to a population-level understanding of suicide dynamics but should not be interpreted as tools for individual-level prediction or surveillance. This distinction is now consistently maintained throughout the Discussion sections (highlighted, page 8 lines 16–19; page 11 lines 19–23; page 13 line 21 – page 14 line 2; page 14 lines 5–10).

Reviewer #2: Thank you for the opportunity to review this paper on the correlation between suicide and suicide-related online searches in Korea. There are two main concerns regarding this paper that I believe should be addressed before the manuscript could be accepted for publication.

Response: We thank Reviewer #2 for the constructive feedback. We have carefully addressed all suggestions to strengthen the manuscript.

1. Overstating claims

In the discussion, many claims and interpretations are overstated. I highlight a few here:

“Collectively, these findings indicate that suicide-related internet search activity often occurs contemporaneously or at short lags relative to suicide rates, underscoring the importance of rigorous statistical approaches, including prewhitening, to ensure validity.” I am not sure how you came to this conclusion?

Response: Thank you for this feedback. We clarify that while previous studies have suggested likely temporal relationships between internet search activity and suicide attempts, evidence across studies remains heterogeneous and inconsistent. This empirical inconsistency necessitates rigorous statistical methods, such as prewhitening, to validate associations and minimize spurious correlations. We have revised the statement to reflect this reasoning more accurately (highlighted, page 9 lines 9–14).

“Search terms were systematically selected to capture key aspects of suicide epidemiology in Korea, including common causes, prevalent methods, major prevention resources, and terms examined in previous research. “ This is a potential limitation and it needs to be addressed more thoroughly than you currently do in the limitation section. The current opinion in the field of suicidality is that involving people with lived experience is necessary; why has this not been done? Also please provide details on how the search terms were systematically selected.

Response: We thank the reviewer for this important point. We agree that involving individuals with lived experience is a best practice in suicidology. However, we clarify two critical distinctions regarding our study design. First, this is an ecological study analyzing aggregate-level data, distinct from person-level research that requires the direct participation of vulnerable individuals. Second, our search term selection was systematic and rigorous, guided by prior literature, the clinical expertise of our investigative team, including psychiatrists and specialists in Korean suicide epidemiology, and established epidemiological dimensions of suicide. We have updated the Methods section to detail this selection process and expanded the Limitations section to explicitly acknowledge the constraints of using predefined terms without direct input from those with lived experience (highlighted, page 3 line 22 – page 4 line 2; page 13 lines 11–15).

You mention “strong contemporaneous correlation” Was it? 0.25-0.26 is moderate at best.

Response: We agree that our terminology should be more precise. While a contemporaneous correlation coefficient of 0.25–0.26 for suicide prevention–related terms is moderate in absolute terms, it was relatively higher than the correlations observed for other search terms in our analysis. We have revised the statement to reflect this relative comparison rather than characterizing it as strong in an absolute sense (highlighted, page 10 lines 4–6). Previous studies examining associations between suicide-related internet search volumes and population suicide rates have reported similarly moderate effect sizes. Our findings thus make a meaningful contribution by demonstrating robust associations for prevention-focused search terms when rigorous statistical methods are applied.

“In contrast, a study from California and Arizona reported that prevention-related searches were followed by declines in emergency department visits for suicidality [27].” I do not understand what is that finding in contrast with?

Response: Thank you for this comment. The contrast refers to the direction and implications of the associations. In our study, prevention-related searches showed a positive contemporaneous correlation with suicide rates, indicating that increased help-seeking occurred alongside elevated suicide rates without a corresponding reduction in suicides. In contrast, the California and Arizona study found that prevention-related searches were followed by declines in emergency department visits for suicidality, suggesting that help-seeking translated into effective intervention and reduced acute suicidal crises. To address your comment, we have revised the manuscript to explicitly highlight this contrast. The updated text now clearly distinguishes the positive association observed in our study from the protective outcomes reported in the U.S. research (highlighted, page 10 lines 15–18).

Not sure if prewhitenting is such a revolutionary methodological approach. It has been done in quite a few papers on this topic, most recently Onie at al (2025) and Colbeth et al (2024).

Response: Thank you for this comment. We clarify that we do not claim prewhitening as a novel methodology. Rather, we emphasize its necessity as a rigorous statistical standard for addressing the non-stationarity and seasonality inherent in suicide time-series data, which are otherwise prone to spurious correlations. Our approach aligns with established best practices and recent literature (e.g., Lee et al., 2020; Tran et al., 2017), where prewhitening has been shown to reduce false positives by filtering out autocorrelation. We have revised the Discussion to explicitly position our use of prewhitening within this existing methodological framework, while highlighting our contribution in comprehensively applying this rigorous standard to the Korean dataset to identify robust, rather than spurious, associations (highlighted, page 9 lines 20–22).

2. Methodological issues

While authors did employ correct statistics, there are a couple more things to address before the approach could be considered rigorous.

a) It is unclear if the Bonferroni correction of p-values was applied to ccf too? Based on the figures, I would guess not.

Response: We confirm that Bonferroni correction was applied to the Cross-Correlation Function (CCF) significance testing. The significance threshold for each CCF value was calculated using the Bonferroni-adjusted alpha level (alpha = 0.05/50 = 0.001) and the effective sample size of the prewhitened series (Threshold = z_{1-\alpha/2}/\sqrt{n}). CCF values exceeding this threshold (in absolute value) are considered statistically significant, as indicated by the significance threshold lines in the figures. We have clarified this procedure in the Methods section.

b) Why did you not examine negative lags? -8 to +8?

Response: We focused on positive lags (0 to 8 weeks) to assess temporal associations in which internet search behavior precedes or is contemporaneous with suicide rates. Negative lags, which would indicate suicide rates preceding search volumes, fall outside the theoretical framework of this study. Our hypothesis posits that individuals search for suicide-related information in response to or concurrently with suicide risk, not that suicide outcomes would precede search behavior. This approach aligns with prior studies examining associations between suicide-related internet searches and suicide rates, which similarly focused on positive lags. We have clarified this rationale in the Methods section (highlighted, page 5 lines 11–12).

c) How were missing values from the Naver Data Lab handled? Were there any missing values?

Response: Naver DataLab provides aggregated weekly search volume data for specified search terms, similar to Google Trends. The platform delivers complete weekly time-series data without missing values for each search term queried. Since we obtained pre-aggregated weekly search volumes for all 25 search terms across the entire study periods, there were no missing values in the search volume data.

d) Please include how the Naver Data was obtained, processed, and aggregated. Unless you do so, your claim that the data is publicly available is not true as researchers cannot replicate your analysis.

Response: Thank you for raising this important point regarding reproducibility. We have revised the Methods section to provide detailed information on data acquisition (highlighted, page 4 lines 8–15). Specifically, we now include: (1) the Naver DataLab URL for direct access, (2) the exact time periods queried (January 2016–December 2019 and January 2020–December 2023), (3) platform settings (mobile and desktop combined), (4) demographic parameters (all ages and both genders), and (5) the normalization method applied by Naver DataLab. This process is analogous to Google Trends: users enter search terms, specify the date range and settings, and receive normalized weekly search volume time-series data. This revised description enables full replication of our data acquisition procedure.

e) In your introduction you have correctly identified the common focus on adolescents in similar research. I was surprised that you did not follow that up in your analysis? Who were the people who died by suicide in your analysis? Is it possible to delineate the analysis by age, even binary one such as adolescents/adults?

Response: We appreciate this thoughtful question and the connection to prior adolescent-focused research. Our study utilizes aggregate national suicide data encompassing all age groups to identify population-wide patterns, deliberately distinguishing it from recent studies specifically focused on adolescents (Choi et al., 2023; Son et al., 2023). By examining general, commonly used search terms across the entire population, we aim to complement, rather than replicate, existing age-specific literature. While age-stratified data are available, conducting parallel analyses would duplicate recent findings without aligning with our primary objective of characterizing broad, population-level associations. We have added a statement to the Limitations section acknowledging that our use of aggregated national data precludes specifying the age distribution of individuals who died by suicide, an inherent constraint of the ecological study design (highlighted, page 13 line 21 – page 14 line 2).

f) Figures are of low quality/resolution

Response: Thank you for this comment. We have improved the figure quality by increasing resolution, adjusting file formats, and optimizing figure dimensions. We have made every effort to balance the complexity of presenting multiple time-series plots with the need for clarity and readability.

g) NB rescaling suici

---

## [Decision Letter · Decision Letter 1]

11 Jan 2026

Temporal relationship of suicide-related internet searches and suicide rates in Korea: a prewhitened cross-correlation analysis

PONE-D-25-47669R1

Dear Dr. Kim,

We’re pleased to inform you that your manuscript has been judged scientifically suitable for publication and will be formally accepted for publication once it meets all outstanding technical requirements.

Kind regards,

Diego A. Forero, MD; PhD

Academic Editor

PLOS One

Additional Editor Comments (optional):

Reviewers' comments:

Reviewer's Responses to Questions

**Comments to the Author**

Reviewer #1: (No Response)

Reviewer #2: All comments have been addressed

2. Is the manuscript technically sound, and do the data support the conclusions?

Reviewer #1: Yes

Reviewer #2: Yes

3. Has the statistical analysis been performed appropriately and rigorously?

Reviewer #1: Yes

Reviewer #2: Yes

4. Have the authors made all data underlying the findings in their manuscript fully available?

Reviewer #1: Yes

Reviewer #2: Yes

5. Is the manuscript presented in an intelligible fashion and written in standard English?

Reviewer #1: Yes

Reviewer #2: Yes

Reviewer #1: Thank you for addressing the comments so thoroughly. I have no further recommendations.

Reviewer #2: (No Response)

**Do you want your identity to be public for this peer review?** For information about this choice, including consent withdrawal, please see our Privacy Policy

Reviewer #1: **Yes:** Jacques Eugene Raubenheimer

Reviewer #2: **Yes:** Lana Bojanić

---

## [Editor Report · Acceptance letter]

PONE-D-25-47669R1

PLOS One

Dear Dr. Kim,

I'm pleased to inform you that your manuscript has been deemed suitable for publication in PLOS One. Congratulations! Your manuscript is now being handed over to our production team.

Kind regards,

on behalf of

Dr. Diego A. Forero

Academic Editor

PLOS One